# A Novel Probiotic-Based Oral Vaccine against SARS-CoV-2 Omicron Variant B.1.1.529

**DOI:** 10.3390/ijms241813931

**Published:** 2023-09-11

**Authors:** Eddie Chung Ting Chau, Tsz Ching Kwong, Chun Keung Pang, Lee Tung Chan, Andrew Man Lok Chan, Xiaoqiang Yao, John Siu Lun Tam, Shun Wan Chan, George Pak Heng Leung, William Chi Shing Tai, Yiu Wa Kwan

**Affiliations:** 1School of Biomedical Sciences, The Chinese University of Hong Kong, Hong Kong, China; chungtingchau@cuhk.edu.hk (E.C.T.C.); kwongtc@link.cuhk.edu.hk (T.C.K.); danielpang@link.cuhk.edu.hk (C.K.P.); michellechan.lt@gmail.com (L.T.C.); andrewmchan@cuhk.edu.hk (A.M.L.C.); yao2068@cuhk.edu.hk (X.Y.); 2Department of Applied Biology and Chemical Technology, The Hong Kong Polytechnic University, Hong Kong, China; sltam@connect.polyu.hk (J.S.L.T.); william-cs.tai@polyu.edu.hk (W.C.S.T.); 3Department of Food and Health Sciences, Faculty of Science and Technology, Technological and Higher Education Institute of Hong Kong, Hong Kong, China; swchan@thei.edu.hk; 4Department of Pharmacology and Pharmacy, The University of Hong Kong, Hong Kong, China; gphleung@hku.hk

**Keywords:** Omicron, *Lactobacillus casei*, oral vaccine, infectious disease, COVID-19

## Abstract

COVID-19 pandemic, caused by the SARS-CoV-2 virus, is still affecting the entire world via the rapid emergence of new contagious variants. Vaccination remains the most effective prevention strategy for viral infection, yet not all countries have sufficient access to vaccines due to limitations in manufacturing and transportation. Thus, there is an urgent need to develop an easy-to-use, safe, and low-cost vaccination approach. Genetically modified microorganisms, especially probiotics, are now commonly recognized as attractive vehicles for delivering bioactive molecules via oral and mucosal routes. In this study, *Lactobacillus casei* has been selected as the oral vaccine candidate based on its’ natural immunoadjuvant properties and the ability to resist acidic gastric environment, to express antigens of SARS-CoV-2 Omicron variant B.1.1.529 with B-cell and T-cell epitopes. This newly developed vaccine, OMGVac, was shown to elicit a robust IgG systemic immune response against the spike protein of Omicron variant B.1.1.529 in Golden Syrian hamsters. No adverse effects were found throughout this study, and the overall safety was evaluated in terms of physiological and histopathological examinations of different organs harvested. In addition, this study illustrated the use of the recombinant probiotic as a live delivery vector in the initiation of systemic immunity, which shed light on the future development of next-generation vaccines to combat emerging infectious diseases.

## 1. Introduction

Coronavirus Disease 2019 (COVID-19) has been a global pandemic since mid-December 2019, and the responsible pathogen was discovered to be a novel coronavirus named SARS-CoV-2 [1]. Since its first emergence, COVID-19 has drastically altered the global public health system, with positive cases reported on all continents [2,3]. The World Health Organization (WHO) estimated that a minimum of 70% of the world population receiving full vaccination is required to stop the pandemic [4]. While the continual evolution of the SARS-CoV-2 virus led to a divergence of new strains with higher transmissibility and lethality, the Omicron parent lineage B.1.1.529 and its descendant lineages remain the currently circulating variants of concern (VOCs) and variants of interest (VOIs) by far [5]. The apparent ability to escape from immune surveillance of the Omicron variants resulted in little or no protection provided by two-dose vaccination despite the short-lasting protection offered by a third booster dose [6]. Most available COVID-19 vaccines were developed based on the ancestral strain of SARS-CoV-2, except the recently WHO-approved Omicron-specific vaccine Comirnaty Original/Omicron BA.1 and Original/Omicron BA.4-5 developed by Pfizer-BioNTech [7]. Thus, there is an urgent need to develop novel preventive measures, particularly targeting the Omicron variants, to prevent the spread of COVID-19 globally.

Mucosal delivery of SARS-CoV-2 antigens is an alluring and compelling strategy to stop COVID-19 spread as the main routes of SARS-CoV-2 transmission in humans are upper respiratory and/or gastrointestinal tracts. Recent efforts have been made for the development of oral vaccines against SARS-CoV-2, such as the *C. reinhardtii* microalgae vaccine [8,9]. Apart from that, Lactic acid bacteria (LAB) is a potential candidate. LAB is a group of widely used Gram-positive bacteria in industrial food fermentation processes. They are regarded as Generally Recognized As Safe (GRAS) by the Food and Drug Administration (FDA) of the United States [10]. While consuming live microorganisms in sufficient quantities, they can survive in the host digestive system and produce several advantageous effects on the host, such as providing anti-colorectal cancer effects and preventing inflammatory bowel diseases [11,12], and are commonly referred to as probiotics [13]. In comparison to traditional injectable vaccines, the administration of probiotic-based mucosal vaccines is easier, safer, and cheaper than traditional injectable vaccines, making them advantageous for mass immunization campaigns, particularly in resource-limited countries [14,15]. In addition to their natural adjuvant and immunomodulatory capabilities, *Lactobacillus casei* has been suggested as one of the potential mucosal vaccine candidates with superior antigen delivery compared to other LABs [16,17]. Recent attempts have been made to deliver SARS-CoV-2 antigens to mucosal surfaces and evoke a protective immune response [18,19]; yet, none of these studies specifically targeted the immune escape variants of SARS-CoV-2.

In this study, a novel oral recombinant *Lactobacillus casei* vaccine expressing the Omicron B.1.1.529 antigens with B-cell and T-cell epitopes, namely OMGVac, was thoroughly validated on its growth kinetics, protein expression, and plasmid stability. For the in vivo study, serum immunoglobulin G (IgG) and serum immunoglobulin A (IgA) levels were detected by the Enzyme-Linked Immunosorbent Assay (ELISA) after oral vaccination of OMGVac to the Golden Syrian hamsters. Upon completion of the immunization regimens, overall safety and toxicology were examined in terms of liver damage markers, histopathology, body weight change, and viscera index. To our knowledge, this is the first study to illustrate the immunomodulatory effects of a probiotic-based vaccine against the SARS-CoV-2 Omicron B.1.1.529 variant. Taken together, this study demonstrated the potential of using live *Lactobacillus casei* as a mucosal vaccine delivery vehicle for triggering immunity against viral pathogens.

## 2. Results

### 2.1. Concept of Study and Construction of OMGVac

This study aims to demonstrate the proof-of-concept of using probiotic *Lactobacillus casei* as an in situ live delivery system of antigenic proteins to the hosts’ mucosal surfaces (i.e., the gastrointestinal tract), consequently provoking an immune response to prevent or counteract the SARS-CoV-2 viral infection (Figure 1). When the recombinant *Lactobacillus casei* that carries the antigen is ingested by the hosts, it will first pass through the stomach, where gastric juice and protein digestive enzymes are present. The thick cell wall of *Lactobacillus casei* is highly resistant to this harsh environment, which can protect the antigen from reaching the intestine [20]. The antigen is hypothesized to be released from the bacterial cell and recognized by the dendritic cells in the intestinal mucosal surfaces, thus triggering the immune response. In addition, *Lactobacillus casei* consists of the peptidoglycan cell wall, a natural immunoadjuvant, which stimulates the immune cells to express specific cytokines to further activate the immune system [21].

To construct the vaccine, a gene insert comprising the secretory signal peptide Usp45, Receptor Binding Motif (RBM), B-cell, and T-cell epitopes of SARS-CoV-2 Omicron B.1.1.529 variant was designed (Figure 1) and codon-optimized before cloning into a highly efficient expression plasmid pPCT2. Thereafter, plasmid pPCT2-Usp45-RBM-B-T-DC was electroporated into competent *Lactobacillus casei* cells and grown in a selective MRS medium for selection. A positive recombinant *Lactobacillus casei* colony, designated as OMGVac, was selected and verified by colony PCR and sequencing analysis before subsequent validation. As a result, a verified OMGVac was successfully constructed.

### 2.2. Validation of Recombinant Protein Expression and Plasmid Stability

Growth kinetics at 37 °C of the recombinant strain was evaluated by using an automated spectrophotometer (BioArchitec, Hong Kong SAR) (Figure 2A). OMGVac strain started to reach its maximum growth rate after 16 h (Log phase) and entered the stationary phase after 40 h. The doubling time of OMGVac was 3.61 h. To evaluate and confirm the expression of recombinant protein in the recombinant strain, levels of the recombinant protein harvested at various time points of OMGVac culture were analyzed by Western blot analysis. Using an anti-6-His tag rabbit polyclonal antibody as the primary antibody, a 34.8 kDa band could be detected, indicating that the recombinant protein was successfully expressed in OMGVac, and protein expression reached its maximum after 24 h of culture (Figure 2B).

In addition, the recombinant strain underwent ten generations of passage to further evaluate the stability of pPCT2-Usp45-RBM-B-T-DC. By polymerase chain reaction (PCR) using the primers Sal1F and OMR, the presence of the 1500 bp band throughout all ten generations confirmed the stability of pPCT2-Usp45-RBM-B-T-DC in OMGVac (Figure 2C).

To round up, 24 h had been selected as the optimal harvest time point for culturing OMGVac with the strongest protein expression. Furthermore, the expression of pPCT2-Usp45-RBM-B-T-DC was shown to be stable in OMGVac for at least 10 generations, which illustrated the plasmid suitability for antigens’ expression of OMGVac.

### 2.3. OMGVac-Induced Immune Response after Two-Dose Regimen

To examine the immunogenicity of OMGVac, a two-dose immunization regimen for 6 weeks was orally administered to Golden Syrian hamsters (Figure 3A). A total of 18 Golden Syrian hamsters were randomly assigned into three groups (*n* = 6): Wild Type; OMGVac; and Control. Wild Type and OMGVac groups received 10 billion CFU/mL of wild-type *Lactobacillus casei* and OMGVac per dose, respectively; the same volume of Phosphate Buffered Saline (PBS) was administered to the Control group. Serum was collected from all animals at 0, 14, 27, and 42 days. IgG and IgA titers in serum were determined by ELISA against SARS-CoV-2 Spike_RBD_ (Omicron B.1.1.529 strain).

The titer of anti-Spike_RBD_ (Omicron B.1.1.529 strain) antibodies was increased from days 0 to days 42 (Figure 3B). Immunization with one dose (Day 14) did not elicit significant IgG antibodies in the OMGVac group when compared to Wild Type and Control groups (Figure 3C). Two weeks after the first immunization (Day 27), the level of IgG of the hamsters that received the OMGVac was significantly higher than that of the Control group (*p* < 0.05) and Wild Type group (*p* < 0.01), respectively (Figure 3D). Moreover, the level of anti-Spike_RBD_ (Omicron B.1.1.529 strain) IgG in the OMGVac group reached the maximum mean titer of 3526 U/mL, which was significantly higher than that of the Control group (*p* < 0.001) and Wild Type group (*p* < 0.001) after receiving secondary immunization (Day 42) (Figure 3E). Therefore, immunization with two doses of OMGVac was required to maintain high titers of IgG antibodies against the Omicron B.1.1.529 variant. However, no significant increase in serum anti-Spike_RBD_ IgA level was detected in vaccinated hamsters (Appendix A).

Altogether, OMGVac was able to elicit IgG systemic immune response in hamsters against the SARS-CoV-2 Omicron B.1.1.529 variant significantly when compared to other groups. A gradual increase in anti-Spike_RBD_ (Omicron B.1.1.529) IgG was observed from days 0 to days 42, reaching its maximum after the two-dose immunization regimen.

### 2.4. Safe Administration of OMGVac Based on Histo- and Physio-Pathological Finding

During the entire study period, no adverse effect in the vaccinated hamsters was observed by the regular veterinarian check-ups. Biological samples at regular intervals from all groups of vaccinated hamsters were collected to evaluate critical aspects of the immune system and assess whether there were any adverse effects. There was no significant difference in body weight change among all groups after two-dose administration on days 0, 14, and 27 (*p* > 0.05) (Figure 4A), except that there was a significant difference on days 42 between the PBS group and the Wild Type group (*p* < 0.05). By the end of the study period, all hamsters were sacrificed, and major organs (spleen, liver, kidneys) were harvested for further assessment. The viscera index of major organs (spleen, liver, kidneys) showed no significant difference among all groups after 42 days of administration (Figure 4B). Furthermore, no significant difference was noticed in the serum level of two liver damage markers, Alanine Aminotransferase (ALT1) (Figure 4C) and Glutamate Oxaloacetate Transaminase (GOT1) (Figure 4D), among all groups. An inadequate serum sample resulted in the inability to obtain serum ALT1 data from one hamster in the Wild Type group. In addition, no specific pathological changes were observed in the liver and spleen of immunized hamsters when compared to the two other groups (Figure 4E).

To sum up, OMGVac was proven to be safe for oral consumption without triggering any adverse effect or change in body weight during the regimen, as evidenced by the post-mortem assessments of the viscera index, level of the liver injury markers, and histopathology of the liver and kidney of the vaccinated hamsters.

## 3. Discussion

SARS-CoV-2, a hypermutated virus with the rapid emergence of new contagious variants, remains the major public health threat in the world after 3 years of circulation within the global population. At present, Omicron is dominant regionally and globally, outcompeting other variants, and harbors multiple amino acid mutations in spikes within the Receptor Binding Domain (RBD) [22]. Despite the high rate of full-dose vaccination in most of the developed countries, the scale-up of vaccination coverage in the resource-limited countries is the bottleneck and remains one of the unmet targets to combat this highly contagious disease. Thus, there is an urgent need to develop an easy-to-use, safe, and low-cost vaccination approach.

Here, we attempted to evaluate the immunogenicity and safety of a novel probiotic-based oral vaccine, OMGVac. With the incorporation of the protein expression vector pPCT2-Usp45-RBM-B-T-DC, recombinant *Lactobacillus casei* successfully expressed the Receptor Binding Motif (RBM), B-cell, and T-cell epitopes of the Spike_RBD_ protein of Omicron B.1.1.529 variant, as clearly demonstrated by Western blot analysis. OMGVac also passed the stability testing as the pPCT2-Usp45-RBM-B-T-DC plasmid remained stable after ten generations of passage. Furthermore, it was revealed that the recombinant probiotic reached its maximum growth, as well as the peak of recombinant protein expression, by 24 h of culture at 37 °C. Based on the growth kinetics and protein expression profile, OMGVac was grown and harvested at the mid-log phase (i.e., approximately 24 h culture) before being subjected to oral administration to ensure the maximum bacterial viability for prolonged survival and optimal expression of antigens for triggering the host immune response. Using the Golden Syrian hamster as an in vivo model, serum ELISA data showed that OMGVac could elicit a high titer of IgG antibodies against the Spike_RBD_ protein of Omicron B.1.1.529 variant after two-dose vaccination, compared with Wild Type and Control groups. Also, the serum IgG level showed an increasing trend from day 0 to day 42 within the same treatment group. Moreover, the safety of vaccine administration was supported by no significant differences in levels of serum liver damage markers (ALT1 and GOT1), body weight, viscera index, and histopathological examinations among all groups of vaccinated hamsters.

Over the past two decades, LAB-based mucosal vaccination has been recommended as one of the most effective approaches to control and prevent emerging respiratory viral infections. Indeed, FAO/WHO recommended various probiotic LAB species, such as *Lactobacillus acidophilus*, *Lactobacillus casei*, and *Lactobacillus rhamnosus* [23], and have become a focus of interest for use in humans as live delivery vehicles for introducing therapeutic and prophylactic molecules directly at the mucosal surfaces [14,24,25,26]. Food-grade LAB can survive passage through, colonize, and remain viable for a period of time in the gastrointestinal tract (GIT) [27]. Unlike Gram-negative bacteria such as *Escherichia coli*, the absence of lipopolysaccharides (LPSs) in cell walls makes it a well-suited candidate for oral administration without the risk of endotoxic shock [28]. On the contrary, the peptidoglycan cell wall of LAB enables it as a natural immunostimulant and adjuvant, which possesses a pathogen-associated molecular pattern (PAMP) that can be detected by the pattern recognition receptors of the host [29]. Pattern recognition receptors, such as Toll-like receptors, are responsible for the initiation of downstream NF-kB and MAPKs signal transduction pathways, which further complements its superior inducer ability of host immune responses [30]. Recent studies on oral and/or nasal administration of recombinant *Lactobacillus casei* showed an efficient induction of protective immunity against various viral infections with superior IgG and IgA levels [31,32,33].

In this study, OMGVac used probiotic *Lactobacillus casei* as an in situ delivery vehicle to deliver antigenic proteins of the SARS-CoV-2 Spike_RBD_ protein of Omicron B.1.1.529 variant into the hosts’ small intestine and stimulate the immune system, especially in the B- and T-lymphocytes-rich Peyer’s patch [34]. When OMGVac passes through the microfold cells in the Peyer’s patch, antigenic protein of the SARS-CoV-2 Spike_RBD_ protein of Omicron B.1.1.529 variant is detected by an antigen-presenting cell. Thus, the T- and B- lymphocytes are stimulated, which gives rise to both IgA and IgG antibody production [35,36]. Of note, mucosal IgA serves as the major neutralizing antibody at the viral-entry mucosal surfaces [37] and shows a higher antiviral effect than IgG [38,39], suggesting that mucosal immunization may elicit superior protective immunity against SARS-CoV-2. Moreover, recent evidence has also suggested that the SARS-CoV2-specific CD4^+^ and CD8^+^ T cells can protect the experimental mice from infection, even in the absence of neutralizing antibodies [40]. Nevertheless, recent studies [41,42,43] attempted to combine different types of antigenic peptides with the antigen, of which dendritic cell peptides have proven to upregulate the antigen’s antigenicity in dendritic cells [44]. Considered collectively, this may hint at the importance of virus-specific T lymphocytes, together with antibodies produced by B lymphocytes, that reinforce the protective effects of OMGVac against the SARS-CoV-2 viral infection. Importantly, the systemic immune response triggered by OMGVac was successfully demonstrated by the significant elevation of virus-specific serum IgG levels in the vaccinated hamsters. However, further study on the mucosal immunity of the vaccinated animal model is warranted as there are no standardized hamster-specific reagents (such as monoclonal antibodies) readily available by the end of this study. Nevertheless, further investigation of the neutralizing and anti-viral efficacy of OMGVac in hamsters by using the SARS-CoV-2 Omicron B.1.1.529 variant in biosafety level 3 facilities is needed. While mice are resistant to SARS-CoV-2 infection due to phylogenetic differences in ACE2 [45,46], Syrian hamsters prevail as the most suitable animal model for studying the immunogenicity and disease pathology of SARS-CoV-2 [47].

To date, numerous injection-based vaccination strategies against SARS-CoV-2 have been developed and yielded remarkable protective effects observed in clinical trials. However, questions remain unanswered regarding the long-term safety use of these vaccines and halted global vaccination coverage. Moreover, currently approved vaccines (e.g., mRNA or inactivated vaccines) require strict cold storage and sophisticated manufacturing capacity. This complicates the distribution of COVID-19 vaccines, especially in less developed countries, resulting in vaccine inequality across different geographical regions. Our oral vaccine, OMGVac, tolerates and survives the lyophilization procedure. It can be stored at room temperature without a trade-off for its efficacy, which exhibits an obvious advantage in terms of reduced costs involved in manufacturing, storage, transportation, and trained medical personnel needed for immunization. It is anticipated that our research findings will benefit the global communities irrespective of the wealth of different geographical regions. Among all the vaccines against COVID-19, OMGVac will be the cheapest in terms of production cost. Taking into account the ease in scale-up production and long shelf-life, everyone can easily consume our oral vaccine at home safely, which brings hope to ease the unfair vaccine inequality observed in the world nowadays.

Certainly, more research is needed to demonstrate the full potential of recombinant probiotics as in situ vaccine delivery vehicles, such as the estimation of mucosal IgA levels in different animal models, assessment of the role of T cells in protection from SARS-CoV-2 viral challenge, and confirmation of the OMGVac’s protection ability to against the current circulating strains of the Omicron variants EG.5, XBB.1.16 and XBB.1.5. Overall, this study provides a novel insight into the use of probiotic-based vaccines against viral infections with solid evidence of the causative agent SARS-CoV-2 Omicron B.1.1.529 variant in the recent COVID-19 pandemic. Once there is a thorough understanding of the mucosal and systemic immunity triggered by probiotic-based oral vaccines, clinical studies can be made possible to further support this non-invasive, low-cost, safe, and effective vaccine strategy to combat emerging infectious diseases.

## 4. Materials and Methods

### 4.1. Ethics Statement and Animals

The present study aims to study the immunogenicity and safety of the oral recombinant probiotic-based vaccine to protect against the SARS-CoV-2 viral infection. The animal model used and procedures involved in this study were approved by the Animal Experimentation Ethics Committee (AEEC) of the Chinese University of Hong Kong (CUHK) (Reference no: 20/146/HMF) and the Department of Health of the Hong Kong Special Administrative Region Government (Date of approval: 10 September 2021). Golden Syrian hamsters used in this study were purchased from and housed in the Laboratory Animal Services Centre, CUHK, with free access to water and food at all times.

### 4.2. Visualization and Statistical Analysis

All the statistical analysis and visualization of results were conducted by GraphPad Prism Version 5.0 (GraphPad Software, San Diego, CA, USA). Two-way Analysis of Variance (ANOVA) with Bonferroni post-test was applied to all comparisons in vivo study (except One-way ANOVA was applied in analyzing ALT1 and GOT1 levels in hamsters’ serum and viscera index). All values were expressed as mean ± standard error of the mean (SEM), and *p* < 0.05 was considered statistically significant unless otherwise specified.

### 4.3. Construction of pPCT2-Usp45-RBM-B-T-DC and OMGVac

The gene insert Usp45-RBM-B-T-DC consists of the sequences encoding the receptor binding motif (a.a. 319-541) [48], B-cell epitope (a.a. 6-31) [49], and T-cell epitope (a.a. 691-699) [50], within the spike protein of the SARS-CoV-2 Omicron strain B.1.1.529 (Figure 1). It also contains a hexahistidine (6x-His) tag joined with adjuvant dendritic cell peptide [51] at the C-terminus and Usp45 signal sequence for antigen secretion. Codon-optimized Usp45-RBM-B-T-DC fragment was synthesized and ligated with a universal vector pUC57 (GenScript, Piscataway, NJ, USA). The pUC57-Usp45-RBM-B-T-DC plasmid was directly transformed into *Escherichia coli* TOP10 competent cells and selected on Luria Broth (LB) supplied with Ampicillin. Usp45-RBM-B-T-DC fragment was released with restriction enzymes MluI and BamHI, thus transforming into a *Lactobacillus casei* expression vector pPCT2 (Figure 1). Transformants with pPCT2-Usp45-RBM-B-T-DC were selected on LB with Erythromycin (Ery) and confirmed by colony PCR followed by gel electrophoresis. The recombinant plasmid pPCT2-Usp45-RBM-B-T-DC was then electroporated into *Lactobacillus casei,* as described previously [52]. A recombinant *Lactobacillus casei* clone with pPCT2-Usp45-RBM-B-T-DC, designated as OMGVac, was selected from selective De Man, Rogosa, and Sharpe (MRS) agar. The presence of pPCT2-Usp45-RBM-B-T-DC was confirmed by PCR and Sanger sequencing using primer pair (Sal1F: 5′-CTCTCAAGGGCATCGGTCGA-3′ and OMR: 5′-CGGTTGTCCGGATCCTTAGTGATGGTGATGGTGA-3′).

### 4.4. Growth Kinetics, Protein Expression, and Plasmid Stability of OMGVac

#### 4.4.1. Growth Kinetics, Protein Extraction, and Western Blot Analysis

A starter culture of OMGVac was grown in 50 mL of MRS broth at 37 °C until turbid. OMGVac was subcultured into 500 mL of freshly prepared MRS broth with a final OD_600_ = 0.005 in triplicates. Turbidimetric determination of OMGVac at 37 °C was performed by an automated spectrophotometer (BioArchitec, Hong Kong SAR, China) by OD_600_ measurement at every hour until the cultures reached the plateau phase. The OD_600_ measurements were then subjected to a growth curve plot against culture time. Doubling time (hrs) was calculated with the following equation:In (2)/[[In (OD_600_ #2 − OD_600_ #1)]/(Time point #2 − Time point #1)](1)

For protein expression analysis, 10 mL of OMGVac culture was collected at various time points for every 8 h of culture. All OMGVac culture samples were centrifuged at 5000 rpm at 4 °C for 10 min and washed with PBS twice to remove excess MRS medium. The pellets were resuspended into 1 mL of 4% SDS in Tris-HCl with 1 mM of phenylmethylsulfonyl fluoride (PMSF), and a total bacterial protein was released by ultrasonication under ice bath conditions. The ultrasonication procedure was 30 cycles of ultrasound at 70% power for 30 s and intervals for 30 s. After ultrasonication, the supernatant was collected by centrifugation at 12,000 rpm at 4 °C for 10 min.

The extracted proteins were separated by sodium dodecyl sulfate-polyacrylamide gel electrophoresis (SDS-PAGE) and electroblotted onto a polyvinylidene difluoride (PVDF) membrane. The membrane was then blocked with 5% *w/v* non-fat dry milk for 1 h and was incubated with 1:2000 6x-His tag rabbit polyclonal antibody (Invitrogen, Carlsbad, CA, USA) at 4 °C overnight, with shaking. After that, the membrane was washed with Tris-buffered saline solution with the detergent Tween^®^ 20 (TBST) three times before and after the incubation with 1:5000 horseradish peroxidase (HRP)-conjugated Anti-Mouse IgG Secondary antibody (ThermoFisher, Waltham, MA, USA) for 1 h. The bound antibodies were detected by the usage of Clarity Max ECL Western Blotting Substrate (BioRad, Hercules, CA, USA). The results were visualized by Amersham ImageQuant 800 Imaging System (Cytiva, Marlborough, MA, USA). Relative intensity calculation was performed according to Ohgane K. and Yoshioka H.’s method [53] for a fair comparison of the OMGVac protein expression in different incubation times.
Relative intensity = Band Intensity of OMGVac-His tag/Band intensity of GroEL.

#### 4.4.2. Plasmid Stability Test of OMGVac

Plasmid stability of pPCT2-Usp45-RBM-B-T-DC in OMGVac was evaluated by the passage of 10 generations. After 10 generations of culture in MRS-Ery, 2 mL of OMGVac culture was collected and pre-treated with lysozyme at 37 °C for 30 min. Then, pPCT2-Usp45-RBM-B-T-DC was extracted and purified by the DNA-spin^TM^ Plasmid DNA Purification Kit (iNtRON Biotechnology, Seongnam, Korea) according to the manufacturer’s instructions. A Polymerase chain reaction was performed to test the presence of pPCT2-Usp45-RBM-B-T-DC with primer pair (Sal1F: 5′-CTCTCAAGGGCATCGGTCGA-3′ and OMR: 5′-CGGTTGTCCGGATCCTTAGTGATGGTGATGGTGA-3′) and visualized in 2% agarose gel electrophoresis.

### 4.5. In Vivo Study of OMGVac on Golden Syrian Hamsters

#### 4.5.1. Bacteria Preparation

A starter culture of OMGVac/Wild Type *L. casei* was grown in 50 mL of MRS broth at 37 °C until turbid. OMGVac/Wild Type *L. casei* were subcultured into 500 mL of freshly prepared MRS broth with a final OD_600_ = 0.005. After 24 h of incubation at 37 °C, the culture was harvested and centrifuged at 12,000 rpm for 10 min, followed by PBS wash 2 times. An equal volume of 50% glycerol in PBS was added to the pellet and resuspended thoroughly. The resulting mixture was aliquoted into 1.5 mL tubes and stored in a −80 °C freezer before use. Plate counting of each aliquot was performed in duplicates to calculate the volume of bacteria required for animal feeding.

#### 4.5.2. Immunization and Sample Collection

Eighteen 5-week-old Golden Syrian hamsters were randomly assigned into 3 groups (*n* = 6 for each group): Wild Type; OMGVac; and Control. During the 6-week study period, a two-dose immunization regimen via oral gavage was applied (Figure 2A). Primary immunization was performed on days 1, 2, and 3, whereas secondary immunization was performed on days 28, 29, and 30. Wild Type and OMGVac groups received 10 billion CFU/mL of wild-type *Lactobacillus casei* or OMGVac per dose, respectively; the same volume of PBS was administered to the Control group. On days 0, 14, 27, and 42, body weight measurements and sera were collected for subsequent analysis. All hamsters were sacrificed for physio- and histo-pathological examination by the end of the 6-week study period.

#### 4.5.3. SARS-CoV-2 SpikeRBD Omicron B.1.1.529 Strain-Specific Serum IgG and IgA ELISA

The immune response of hamsters triggered by OMGVac was determined by hamster IgG and IgA enzyme-linked immunosorbent assay (ELISA), as described previously [54]. In brief, 96-well microplates were coated with 2 µg/mL of SARS-CoV-2 Spike_RBD_ Omicron B.1.1.529 strain (GenScript, USA) overnight at 4 °C. After blocking with 5% bovine serum albumin in PBS, 1:100 diluted hamster sera and serial diluted SARS-CoV-2 (Omicron) Neutralizing Antibody Standard (GenScript, USA) were added in triplicates to the microplates and incubated in the dark for at least 1 h for IgG assay. For the IgA assay, 1:50 diluted hamster sera were added in triplicate to the microplates. The IgG and IgA antibodies were detected by using HRP-conjugated goat anti-hamster IgG antibody (Invitrogen, USA) and HRP-conjugated rabbit anti-hamster IgA antibody (Brookwoodmedical, Homewood, AL, USA), respectively. The 3,3′,5,5′-Tetramethylbenzidine (TMB) was used as a substrate. After the addition of 1 M Hydrochloric acid (HCl) for stopping the crosslinking, optical density at 450 nm was measured by an xMark^TM^ Microplate Absorbance Spectrophotometer (BioRad, USA).

#### 4.5.4. Hamster Alanine Aminotransferase 1 (ALT1) and Glutamate Oxaloacetate Transaminase 1 (GOT1) ELISA

Liver damage of OMGVac was evaluated by ALT1 and GOT 1 enzyme-linked immunosorbent assay (ELISA) kits (MyBiosource Inc., San Diego, CA, USA) according to the manufacturer’s protocols. In brief, 50 µL of undiluted hamster sera were added to the pre-coated ALT1 and GOT1 96-well plate. Then, 100 µL of HRP-conjugated reagent was added to two plates, and the plates were incubated in a 37 °C incubator for 60 min. After several washing steps, chromogen solutions A and B were added to each well, and the plates were incubated at 37 °C for 15 min. The stop solution was added to the plates, and the optical density at 450 nm was measured by an xMark^TM^ Microplate Absorbance Spectrophotometer (BioRad, USA).

#### 4.5.5. Physio- and Histo-Pathological Examination

By the end of the study period (Days 42), major organs (liver, spleen, and kidneys) were harvested and weighed. Both liver and spleen tissues were then immediately preserved in 10% neutral buffered formalin (NBF) for no more than 24 h. Well-preserved tissues were processed by the Epredia Excelsior AS Tissue Processor (ThermoScientific, Waltham, MA, USA) and embedded with paraffin wax by the Epredia HistoStar Embedding Center (ThermoScientific, USA). Then, the formalin-fixed paraffin-embedded (FFPE) blocks were cut into 0.5 µm thick sections by a Rotary Microtome (Leica BioSystems, GmbH, Nußloch, Germany) and mounted on glass slides before being subjected to hematoxylin and eosin (H&E) staining. The stained spleen and liver were examined by the Zeiss Axioscan 7 Automatic Slide Scanner with 40× magnification and 20× magnification. The viscera index calculation was described in [18]. The viscera index of the hamster’s major organs (liver, spleen, and kidneys) was calculated with the following equation:

(Weight of Organ/Body weight before sacrifice) × 100

## 5. Conclusions

In summary, our experimental findings demonstrated that OMGVac is a promising candidate to combat the COVID-19 pandemic by eliciting a significant level of IgG immune response in hamsters. Also, no serious adverse effects were observed throughout this study. The vaccine safety was supported by the liver damage marker assessment and physio- and histo-pathological examinations after oral consumption of OMGVac for 42 days. Everything considered, the use of recombinant probiotics as in situ delivery vectors clearly illustrated apparent advantages over traditional vaccination strategies. However, further investigation is required to explore the full scope of neutralizing activity and the underlying mechanism of recombinant probiotics against viral infection.

## Figures and Tables

**Figure 1 ijms-24-13931-f001:**
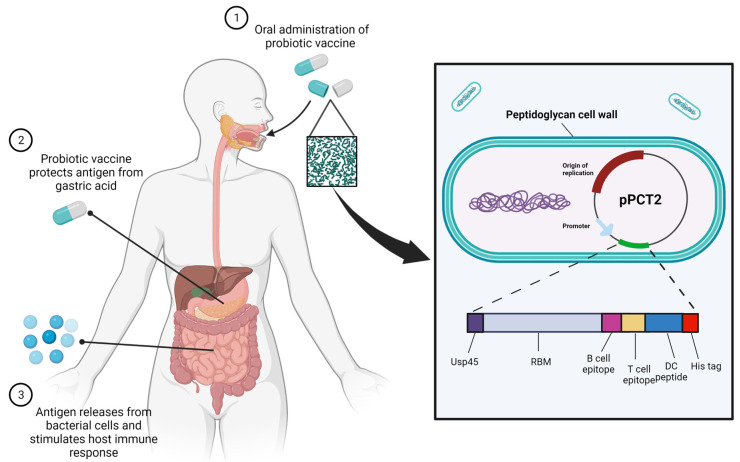
Concept and design of the probiotic-based oral vaccine OMGVac (created with BioRender.com).

**Figure 2 ijms-24-13931-f002:**
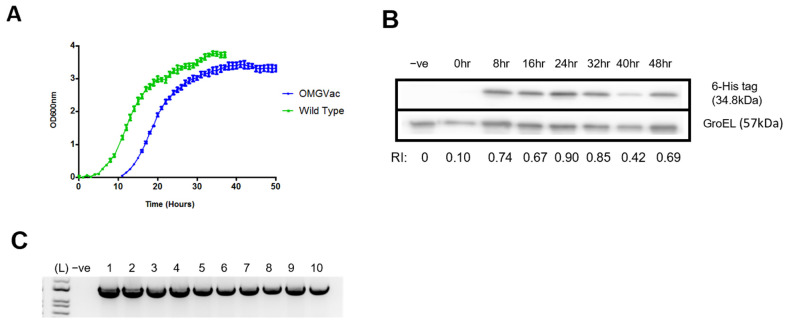
Validation of recombinant protein expression and pPCT2-Usp45-RBM-B-T-DC plasmid stability. (**A**) Growth curve of OMGVac at 37 °C. (**B**) Western blot analysis of recombinant protein across various incubation time points of OMGVac. (RI = Relative intensity of 6-His tag band compared with GroEL band). (**C**) Stability PCR of pPCT2-Usp45-RBM-B-T-DC plasmid over 10 passages of OMGVac culture.

**Figure 3 ijms-24-13931-f003:**
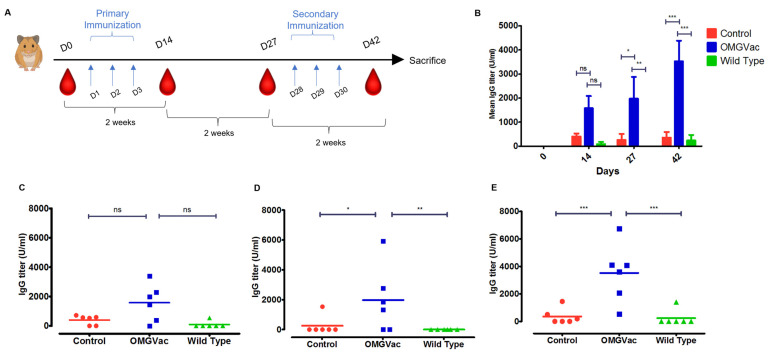
Study overview and effects of OMGVac on Omicron-specific antibody responses in vaccinated hamsters (*n* = 6 per group). ns = not significant, * = *p* < 0.05, ** = *p* < 0.01, *** = *p* < 0.001 (**A**) Study overview and dosing regimen of a 6-week study period (created with BioRender.com). (**B**) Mean serum IgG against Omicron variant B.1.1.529 RBD titer of vaccinated hamsters from 0 to 42. (**C**) Serum IgG against Omicron variant B.1.1.529 RBD titer of vaccinated hamsters after primary immunization (Day 14). (**D**) Serum IgG against Omicron variant B.1.1.529 RBD titer of vaccinated hamsters before secondary immunization (Day 27) (**E**) Serum IgG against Omicron variant B.1.1.529 RBD titer of vaccinated hamsters after secondary immunization (Day 42).

**Figure 4 ijms-24-13931-f004:**
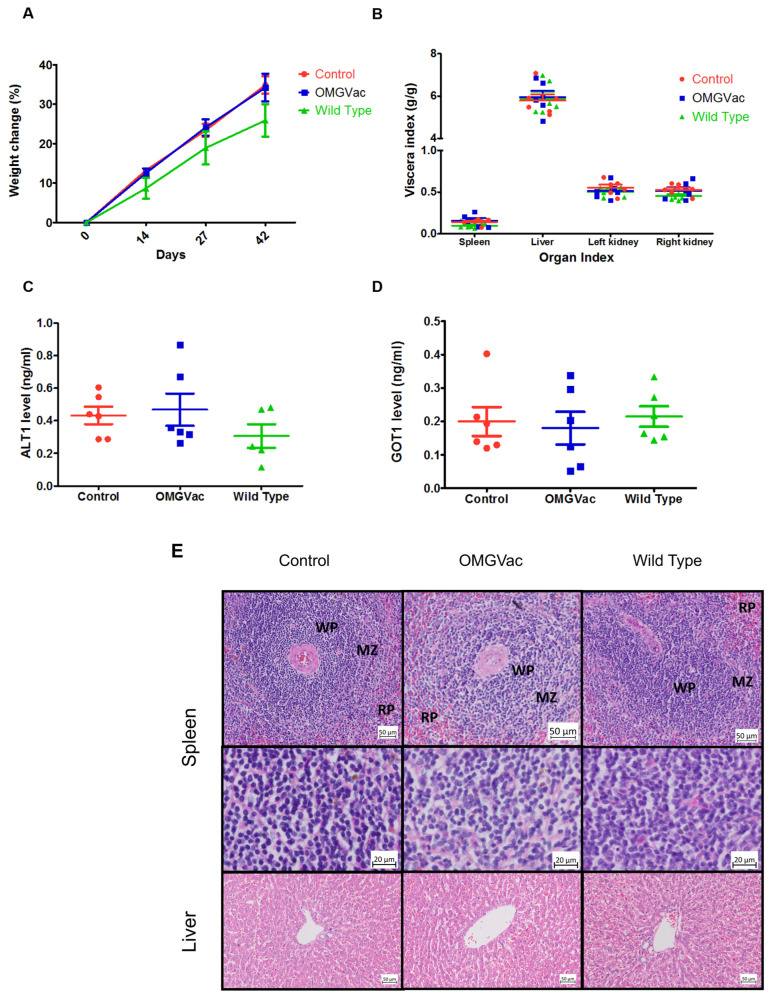
Physio- and Histo-pathological Examination of Vaccinated Hamsters. (**A**) Body weight percentage change in hamsters throughout the experimental period (from day 0 to days 42). No statistical significance was observed (*p* > 0.05), except there was a significant difference on day 42 between Wild Type and PBS Control group (*p* < 0.05). (**B**). Viscera index of different hamsters’ organs in control, OMGVac, and Wild Type group. No statistical significance was observed (*p* > 0.05). (**C**) Serum ALT1 level on day 42. No statistical significance was observed (*p* > 0.05). (**D**) Serum GOT1 level on day 42. No statistical significance was observed (*p* > 0.05). (**E**) Histopathological examination of the spleen (WP = white pulp, MZ = marginal zone, RP = red pulp) and liver from representative hamsters of each group (H&E stain).

## Data Availability

The vector in this study, pPCT2, will be subjected to the patent application; thus, the details of pPCT2 will not be disclosed.

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
