# Peer review of "A Novel Probiotic-Based Oral Vaccine against SARS-CoV-2 Omicron Variant B.1.1.529"

_ijms, 2023, doi:10.3390/ijms241813931_

Round 1

Reviewer 1 Report

In this report, Eddie Chung Ting Chau et al. have developed a novel oral recombinant Lactobacillus casei vaccine expressing the Omicron B.1.1.529 antigens with B-cell and T-cell epitopes, named OMGVac, and studied its immunogenicity and safety in Golden Syrian hamsters after a prime-boost immunization regimen. The authors found that the vaccine was able to elicit high levels of IgG two weeks after first immunization which increased further after boost immunization as compared to the control and wild-type groups. The safety of the vaccine was also assessed by examining liver damage markers, histopathology, body weight changes and viscera index.

This is a very interesting proof-of-concept report studying the immune responses elicited by the novel probiotic-based vaccine (OMGVac) in hamsters after oral immunization. The results from the study clearly indicate that OMGVac can be developed as a promising candidate against COVID-19 after further studies. I found the paper to be very well-written and clear. The experimental layout is methodical. Though there are some concerns about the absence of important results such as serum neutralizing activity data, the authors have highlighted this limitation in the conclusion section.

To improve the quality of this manuscript, below are some comments that need to be addressed and/or answered by the authors.

1.       Please include a conclusive statement at the end of each subsection in the results section summarizing what each figure and data describes.

2.       Figure 3: For Figs. 3B-E, please include the antigen used for ELISA (IgG to Omicron RBD) either on the Y-axis label or preferably as a separate title for each sub-figure. It is also suggested that the authors include the ‘n’ number per group in the figure legends for the Golden Syrian hamsters used in the study.

3.       Since it is very well known that oral vaccination induces high levels of IgA, it is surprising that no significant differences in the levels of serum IgA were found in the vaccinated hamsters. Do the authors have a plausible explanation for why there were no significant differences in serum IgA levels in the vaccinated hamsters?

4.       A limitation of the study is that it is only focused on evaluating one aspect of humoral immunity and does not delve deeper into evaluating the impact of T cell-mediated immunity. In the discussion section, there is a brief mention of the importance of T cells in contributing to protection from SARS-CoV-2 infection. Since the OMGVac expresses the Omicron B.1.1.529 antigens with T cell epitopes, it would be good to highlight the need for further studies assessing the role of T cells in protection from viral challenge in the discussion section of the paper as a future direction that could be explored.

5.    An additional point that should be added to the discussion section is the need for performing further studies to confirm if OMGVac could potentially be used against the currently circulating strains of Omicron (sub-variants) indicating broader protection.

Very minor editing of English language required.

Reviewer 2 Report

 The manuscript entitled “A Novel Probiotic-based Oral Vaccine Against SARS-CoV-2 2 Omicron Variant B.1.1.529 ” has a novel and original topic that might attract many readers.

The manuscript is designed very well and has a good English style. From my point of view, this manuscript is suitable for publication and I present some comments that might be useful for the authors.

-Please add some sentences about the potential of Lactobacillus species in the treatment/prevention of some infectious diseases like gastric cancer.

-Please highlight the important features of edible vaccines than the other routine vaccines.

-Please add some efforts that have been made in plants and micro-algae about edible vaccine production against the SARS-CoV-2 virus. You can use the flowing reference (https://doi.org/10.3390/md20110657).           
